# Very Short-Term Surface Solar Irradiance Forecasting Based on FengYun-4 Geostationary Satellite

**DOI:** 10.3390/s20092606

**Published:** 2020-05-03

**Authors:** Liwei Yang, Xiaoqing Gao, Jiajia Hua, Pingping Wu, Zhenchao Li, Dongyu Jia

**Affiliations:** 1Key Laboratory of Land Surface Process and Climate Change in Cold and Arid Regions/Northwest Institute of Eco-Environment and Resources, Chinese Academy of Sciences, Lanzhou 730000, China; yanglw@lzb.ac.cn (L.Y.); zhenchaoli@lzb.ac.cn (Z.L.); 2Tangshan Meteorological Service/CMA, Tangshan 063000, China; ustchuajiajia@163.com; 3Key Laboratory of Meteorology and Ecological Environment of Hebei Province, Shijiazhuang 050000, China; 4Weichang Manchu Mongolia Autonmous County Meteorological Bureau/CMA, Chengde 068450, China; wc54311wpp@163.com; 5College of Geography and Environmental Engineering, Lanzhou City University, Lanzhou 730070, China; jdy890719@lzb.ac.cn

**Keywords:** solar energy, FengYun-4A satellite, solar power plant, surface solar irradiance, forecast

## Abstract

An algorithm to forecast very short-term (30–180 min) surface solar irradiance using visible and near infrared channels (AGRI) onboard the FengYun-4A (FY-4A) geostationary satellite was constructed and evaluated in this study. The forecasting products include global horizontal irradiance (GHI) and direct normal irradiance (DNI). The forecast results were validated using data from Chengde Meteorological Observatory for four typical months (October 2018, and January, April, and July 2019), representing the four seasons. Particle Image Velocimetry (PIV) was employed to calculate the cloud motion vector (CMV) field from the satellite images. The forecast results were compared with the smart persistence (SP) model. A seasonal study showed that July and April forecasting is more difficult than during October and January. For GHI forecasting, the algorithm outperformed the SP model for all forecasting horizons and all seasons, with the best result being produced in October; the skill score was greater than 20%. For DNI, the algorithm outperformed the SP model in July and October, with skill scores of about 12% and 11%, respectively. Annual performances were evaluated; the results show that the normalized root mean square error (nRMSE) value of GHI for 30–180 min horizon ranged from 26.78 to 36.84%, the skill score reached a maximum of 20.44% at the 30-min horizon, and the skill scores were all above 0 for all time horizons. For DNI, the maximum skill score was 6.62% at the 180-min horizon. Overall, compared with the SP model, the proposed algorithm is more accurate and reliable for GHI forecasting and slightly better for DNI forecasting.

## 1. Introduction

The rapid growth of global energy demands has posed serious challenges for the realization of sustainable economic and social development after entering the 21st century. As the largest clean and renewable energy source on earth, solar energy is expected to become the largest power source in the world in the future. The cost of solar panels and related devices has declined dramatically, creating the conditions for large-scale research and application of photovoltaic power generation systems in the coming decades [1,2,3]. Photovoltaic arrays mainly use global solar irradiance to generate output power. Because surface solar irradiance is easily affected by meteorological factors (mainly clouds), the photovoltaic output power fluctuates and changes considerably. With increases in the installed power ratio, large fluctuations in the output power have a huge impact on the power grid system, which could have serious and dangerous consequences [4,5]. Therefore, accurate prediction of surface solar irradiance can provide important decision-making support for power dispatching, effectively reducing the operation costs of power systems which depend solely on photovoltaic resources, and produce greater economic and social benefits [6,7].

For different temporal and spatial resolutions of surface solar irradiance predictions, the methods used are also different. Medium- and long-term predictions can be used in the construction and planning of photovoltaic power stations and other fields; climate models from the Coupled Model Intercomparison Project Phase 5 (CMIP5) under different scenarios are most commonly used [8,9]. Short-term predictions can provide hourly variations in surface solar irradiance, and the prediction results can be used in optimal microgrid scheduling. The main algorithms used in this field are numerical weather prediction (NWP) models, statistical methods, and artificial neural networks (ANNs) [10,11,12,13]. Very short-term surface solar irradiance forecasting, also called “now-casting” (0–3 h ahead) can provide information about photovoltaic power variation in the high-frequency range, which is mainly used in transient analyses and control strategy research of micro grids. Ground-based sky images and satellite images are mostly used to obtain the cloud motion vectors (CMV) field [14,15]. When the forecast horizon is in the range of 0–3 h, the CMV-based method performs better than the NWP model and ANN [14,16].

The objective of this study is very short-term forecasting (0–3 h) of surface solar irradiance. The spatial-temporal resolution of ground-based cloud images is very high, but the observation range of the all-sky imager is limited (e.g., the effective monitoring range of TSI-880 is 5 km). For three-hour-ahead forecasting, the cloud cluster may have moved out of the monitoring range of the imager. Therefore, the all-sky imager is generally suitable for 0–15 min-ahead forecasting. As for the limitations of all-sky imager monitor, a larger range image from the satellite needs to be considered for 0–3 h-ahead forecasting. To a user, e.g., a solar power plant, satellite cloud images are accessible immediately and almost for free on the Internet. Thus, satellite cloud images are inexpensive and can accurately reveal the characteristics of regional cloud cover [17,18].

Satellite observation has been used for surface solar irradiance forecasting in different ways. Arbizu-Barrena, et al. [10] used Meteosat Second Generation (MSG) for short-term solar radiation forecasts (up to six hours ahead) of global horizontal irradiance (GHI) and direct normal irradiance (DNI) by forecasting cloud indices using the Weather Research and Forecasting (WRF) numerical weather prediction (NWP) model. Nonnenmacher and Coimbra [19] used satellite images and ground measurements for very short-term GHI forecasts (1–3 h ahead) with an approach based on an optical flow algorithm to track and advect cloud. Wang, et al. [14] used a MSG geostationary satellite for very short-term GHI and DNI forecasts (0–4 h ahead). Their algorithm was developed using the SEVIRI (imager on MSG) cloud physical properties. Gallucci, et al. [20] used MSG data for nowcasting surface solar irradiance (up to two hours in advance); the approach was based on advection and extrapolation of MSG-SEVIRI channels. Ayet and Tandeo [21] used MSG satellite images for nowcasting solar irradiance (up to six hours ahead) of GHI; they statistically emulated cloud dynamics, and no numerical weather model was required. Rosiek, et al. [22] applied MSG satellite observations and ANN for online three-hour forecasting of GHI and the building integrated photovoltaic (BIPV) power output.

The aforementioned studies show that MSG is the most widely-used satellite for short-term solar irradiance forecasting. FengYun-4A (FY-4A) is a new generation of geostationary meteorological satellite for quantitative applications in China. It provides high resolution earth observations of geosynchronous orbit in China and is equipped with an advanced geosynchronous radiation imager (AGRI) which can obtain 14-band channels of the earth’s cloud images at high frequency. Compared with the FengYun-2 radiation imager, AGRI’s observation performance is considerably better [23,24]. FY-4A satellite images have rarely been used for surface solar irradiance nowcasting [25]. Therefore, we conducted this study on surface solar irradiance nowcasting based on FY-4A satellite images, which has profound significance for the system safety of photovoltaic power stations and grids in China.

In this work, we developed and evaluated an algorithm to forecast very short-term (30–180 min) surface solar irradiance using visible and near infrared (AGRI) channels on board the FY-4A geostationary satellite. The forecasting products of surface solar irradiance include GHI and DNI. The European Solar Radiation Atlas (ESRA) model [26] was used to calculate the surface solar irradiance under clear sky conditions. As a solar irradiation attenuation model, we used the Heliosat-2 method [27]. Particle image velocimetry (PIV) was used to obtain the cloud motion vector (CMV) field from the satellite images. The GHI and DNI forecast results were assessed using ground-based measurements at the Weichang meteorological station in Chengde of four typical months (October 2018, and January, April, and July 2019) under all sky conditions. Finally, the FY-4A satellite-based method was compared with the classical smart persistence (SP) model.

In this algorithm, historical data and observed meteorological elements, such as temperature, wind speed, and so on, are not required. Many forecasting methods with high precision are available in this field, but they require more meteorological variables, high quality data, complex calculations, a high performance computer, etc., which is costly and inconvenient for industrial application. Our algorithm has lower computational cost and it is applicable in areas where a meteorological observation network system is not always available.

The paper is structured as follows: Section 2 describes the observation data including satellite observations and ground measurements. In Section 3, we present the surface solar irradiance forecasting algorithm and model performance metrics. In Section 4, the GHI and DNI forecast results are presented and evaluated. In Section 5, the conclusions and discussion are outlined.

## 2. Measurements

### 2.1. Satellite Images

FengYun-4, a new generation of Chinese geostationary meteorological satellite, was successfully launched on 11 December, 2016. It is equipped with an advanced geosynchronous radiation imager (AGRI) which has 14 channels between 0.47 and 13.5 µm; the spatial resolution is 0.5–4 km. The satellite images can be downloaded from: http://satellite.nsmc.org.cn/. Satellite images with a solar elevation angle of less than 12° are not used (i.e., the measurement error of the instrument caused by the shielding of the surrounding mountains and calculation error caused by the low brightness of the satellite images). In this study, the visible (0.63–0.69 µm) and near infrared (0.76–0.9 µm) channels were used for cloud observation (identify cloudy groups for cloud index derivation and CMV calculation). Images from FY-4A were captured from August 2018 to September 2019 with a 2 × 2 km spatial resolution. Figure 1 shows that solar irradiance fluctuates considerably due to the movement of clouds.

### 2.2. Ground-Based Observations

The Beijing-Tianjin-Hebei (BTH) region is an important energy consumption area, accounting for more than 10% of the total energy consumption in China. Due to the increasingly serious pollution problem in this area, developing clean energy has become a key direction for energy transition [28].

The city of Chengde is located in the BTH region. Due to the unique natural and geographical conditions, Chengde has a variety of clean energy types and rich resources, and has become a national new energy demonstration city. The Chengde climate is temperate continental monsoon, with four distinctive seasons. The altitude in Chengde is 200–1200 m. The average annual temperature is 5.6 °C, the average annual precipitation is 300–560 mm, and the average annual sunshine is 2845 h.

The observation data of global horizontal irradiation (GHI), direct normal irradiance (DNI), and diffuse horizontal irradiation (DHI) were provided by the Weichang national basic meteorological station in Chengde (Station No.: 54311). The geographical location is shown in Figure 2 (117°45′58′′ E, 41°57′38′′ N; altitude 892.7 m). The solar irradiance data are measured at a 1-min time resolution. The GHI and DNI are recorded using FS-S6 and FS-D1 solar radiation sensor, respectively. The instruments are calibrated every two years, conforming to the First Class technical indicators of the ISO 9060:1990 and WMO standards.

## 3. The Description of Forecasting Methods

### 3.1. ESRA Model

The accuracy of the clear sky radiative model directly impacts the solar irradiance forecast results. Many validation studies have been conducted to compare the accuracy, advantages, and limitations of each clear sky radiative model [29,30,31,32,33]. In this work, we selected a clear sky model with high precision both for GHI and DNI data and low complexity. After a detailed investigation, we selected the European Solar Radiation Atlas (ESRA) model to calculate the surface solar irradiance. The method details can be found in the literature [26]. The calculation of the Linke turbidity factor (*T*_L_) is the critical input for the accuracy of the model. The *T*_L_ in this work was calculated using the following equation [34]: (1)$$T_{L}=3.91\tau_{550}\exp(0.689\times \frac{1013.25}{p})+0.376\ln(u_{H_{2}O})+[2+0.54\frac{1013.25}{p}-0.5{\left(\frac{1013.25}{p}\right)}^{2}+0.16{\left(\frac{1013.25}{p}\right)}^{3}]$$
where $\tau_{550}$ is the aerosol optical depth (AOD) at 550 nm (dimensionless), $u_{H_{2}O}$ is the atmospheric precipitable water value (pwv; mm), and *p* is the local atmospheric pressure (hPa).

In this work, the pwv data were collected from the Chengde meteorological station (monitoring GPS/MET moisture observation station data); the value acquired was from the most recent time (according to the time horizon) for the present day. The AOD values were acquired from monthly averaged data [35,36,37]; the values for October, January, April, and July were 0.03, 0.04, 0.09, and 0.14, respectively. Heavy pollution days with an air quality index (AQI) greater than 120 μg/m^3^ were not considered. Using this method to calculate *T*_L_, the normalized root mean square deviation (nRMSE) of the ESRA clear sky model was found to be 3.1% for GHI and 5.6% for DNI based on the days under clear sky conditions in this work. For example purposes, the measured and forecast solar irradiances under clear sky condition are presented in Figure 3 for one day each (1 min time resolution) in April and September. The GHI and DNI components were considered in this study.

Better turbidity estimation is crucial for improving the accuracy of the clear sky model. Highly accurate retrieval of aerosol optical thickness (AOD) is the challenge. Retrieving high-resolution AOD using the FY-4A satellite has been researched [38,39,40]. The retrieved high-resolution AOD is very important for the BTH region we studied, as photovoltaic plants are developing rapidly but are also facing problems due to air pollution. As such, retrieving AOD data using the FY-4A satellite to improve the accuracy of the ESRA algorithm will be our next work.

### 3.2. The Heliosat-2 Method

As a solar irradiation attenuation model, we used is the Heliosat-2 method [27]. The model has been applied to study the attenuation of solar irradiation reaching the ground using satellite images, and to forecast the surface solar irradiation [10,22,41,42,43,44]. The general idea of the physical model involves converting the satellite observation pixel image into a cloud index pixel image, and multiplying the cloud index with solar irradiation using the clear sky model. Then, the surface irradiation at any given location (pixels) can be obtained.

The attenuation caused by meteorological factors in the atmosphere is obtained by $K_{c}$, which is calculated using the following formula: (2)$$\left\{ \begin{array}{l} n^{t}(i,j)<-0.2,\;K_{c}=1.2\\ -0.2<n^{t}(i,j)<0.8,\;K_{c}=1-n^{t}(i,j)\\ 0.8<n^{t}(i,j)<1.1,\;K_{c}=2.0667-3.6667n^{t}(i,j)+1.667{\left(n^{t}(i,j)\right)}^{2}\\ n^{t}(i,j)>1.1\;K_{c}=0.05 \end{array}\right.$$
where $K_{c}$ is the clear sky index (dimensionless), and $n^{t}(i,j)$ is the cloud index (dimensionless) at the instant time t and pixel (i,j). Its value is obtained using Equation (3): (3)$$n^{t}(i,j)=\frac{\rho^{t}(i,j)-\rho_{g}{}^{t}(i,j)}{\rho_{c}{}^{t}(i,j)-\rho_{g}{}^{t}(i,j)}$$
where $\rho^{t}(i,j)$ is the apparent albedo (dimensionless), $\rho_{g}{}^{t}(i,j)$ is the apparent albedo of the ground under clear sky conditions (dimensionless), and $\rho_{c}{}^{t}(i,j)$ is the apparent albedo of the brightest clouds (dimensionless). The FY-4A satellite level 1 data provide a digital number which can be directly converted to apparent albedo after looking it up in the calibration table. To find the albedo of the brightest clouds and the ground for each pixel, a histogram of all pixels throughout a year is constructed through statistical analysis. The 95th percentile of the histogram was selected as the albedo of the brightest clouds, while the darkest pixels from its monthly time series were selected as the ground albedo.

The DNI estimation method can be found in [45], which is based on the diffuse fraction model by Skartveit, et al. [46].
(4)$$DNI=DNI_{cs}\times {\left(K_{c}-0.38\times (1-K_{c})\right)}^{2.5}$$

When the clear sky index $K_{c}$ is smaller than 0.35, the DNI value is set to zero.

### 3.3. Forecasting Model Process

The cloud index and clear-sky index pixels were obtained using the Heliosat-2 method. The particle image velocimetry (PIV) algorithm can record a large amount of velocity distribution information at the transient point and provide the spatial structure and flow characteristics of the velocity field. The MPIV is a PIV toolbox written in MATLAB; the detailed method is described in [47], and the full package of the MPIV toolbox can be downloaded free of charge. Here, MPIV was used to calculate the cloud velocity vector for forecasting the clear sky index, which will move to pixels (the area in which we were interested) in the future. To guarantee that the fastest-moving cloud will not move out of the area being studied, the FY-4A satellite cloud image pixel area was selected as 400 × 400 pixels, with the research pixel in the center. The latest two consecutive satellite images were used. In the MPIV method, the area of the square portion needs to be set to find the most similar and the same-sized portion in the continuous image to obtain the cloud motion vector (CMV) of the selected square portion. The corresponding MPIV criterion to search for the best match block was set to the minimum quadratic difference (MDQ) method. In this work, we selected the square portion as 32 × 32 pixels, and overlapping was not allowed. To ensure the accuracy of the results, filtering (global filter) was applied and unrealistic vectors were removed. An interpolation algorithm (weighted averaging of adjacent pixel value) was used to replace the invalid values. Then, the cloud velocity field was obtained. Finally, we obtained a value of the clear sky index for the area of interest with different time horizons. The time horizons in this work were set to 30, 60, 90, 120, 150, and 180 min.

The surface solar irradiances were obtained by multiplying the clear sky index by the clear sky irradiances. The main GHI and DNI forecasting process steps are summarized in Figure 4. The time horizons ranged from 30 to 180 min; for example, a time horizon of 30 min indicates a 30-min-ahead forecasting process. The MATLAB 2016b mathematic software was employed to model the forecasting algorithm.

### 3.4. Performance Metrics

It is meaningless to simply compare models. Each model has its own applicable scenarios, advantages, and disadvantages. To evaluate the quality of the model, it needs to be placed in a specific data scenario. Real data must be used to determine which model is better in an applied scenario. Therefore, common error metrics were employed to evaluate the performance of the solar forecasting algorithm: root mean square error (RMSE), mean absolute error (MAE), mean bias error (MBE), the correlation coefficient (*R*), and the normalized RMSE (nRMSE), MAE (nMAE), and MBE (nMBE), which are defined as [48]: (5)$$RMSE=\sqrt{\frac{1}{N}\times \sum_{i=1}^{N}{\left(y_{i}-x_{i}\right)}^{2}}$$
(6)$$nRMSE=\frac{RMSE}{\bar{x}}\times 100\%$$
(7)$$MAE=\frac{1}{N}\times \sum_{i=1}^{N}\left|y_{i}-x_{i}\right|$$
(8)$$nMAE=\frac{MAE}{\bar{x}}\times 100\%$$
(9)$$MBE=\frac{1}{N}\times \sum_{i=1}^{N}\left(y_{i}-x_{i}\right)$$
(10)$$nMBE=\frac{MBE}{\bar{x}}\times 100\%$$
(11)$$R=\frac{\sum_{i=1}^{N}\left(x_{i}-\bar{x})(y_{i}-\bar{y}\right)}{\sqrt{\sum_{i=1}^{N}\left(x_{i}-\bar{x}\right)}\sqrt{\sum_{i=1}^{N}\left(y_{i}-\bar{y}\right)}}$$
where x is the ground measured value, y is the forecasted value, and *N* is the number of data.

### 3.5. Forecast Skill

We used a new metric, the forecast skill score (SS), which is expressed as [49,50]: (12)$$SS=1-\frac{nRMSE_{forecast}}{nRMSE_{persistence}}$$
where $nRMSE_{forecast}$ is the nRMSE value for the solar forecasting algorithm, and $nRMSE_{persistence}$ is the nRMSE value for the smart persistence (SP) model. The SP model is defined as [51]: (13)$$\overset{\wedge }{y}(t+\tau )=y(t)\times \frac{y^{clr}(t+\tau )}{y^{clr}(t)}$$
where $\overset{\wedge }{y}(t+\tau )$ is the solar irradiance forecasting value at time *t*, y(t) is the ground measured value, $y^{clr}(t)$ is the value of solar irradiance calculated from the clear sky model (ESRA in this work) at time *t*, and τ is the forecast horizon (30–180 min in this work). For the SP model, only the ground measured irradiances and the clear sky irradiances are needed, and the key consideration is the current measured deviation of the irradiances from clear sky irradiances. Although the SP model is simple, it is accurate for the prediction of periods with low GHI (or DNI) variability. This smart persistence model mainly served as a reference model; therefore, the predicted results of any forecast model being better than the SP model indicates improvement in the ability to predict random variability [52].

When the forecast skill score (SS) was 1, the algorithm produced perfect forecast results over the reference model; when the SS was 0, there was no improvement against the reference model; and when the SS was a negative value, the performance of forecast algorithm was worse than the reference.

## 4. Results and Discussion

It’s interesting to analyze the accuracy of the model by season (typical months) under all sky conditions, so as to analyze the impact of the meteorological conditions in four seasons on the performances of the models.

### 4.1. Seasonal Studies

#### 4.1.1. Global Horizontal Solar Irradiation: GHI

The all-sky conditions considered here refer to clear, partly cloudy, and overcast skies. The RMSE is related to the average value of each season. Comparing the accuracy of the model in different seasons using RMSE or MAE is difficult, so we used the nRMSE and nMAE to analyze the performance of the models. Figure 5; Figure 6 shows the nRMSE and nMAE values of the forecasting models for GHI and DNI in four typical months (October 2018; January, April, and July 2019) at the study location. The figures show that the best results for GHI for both models were produced in January for all forecasting horizons, followed by October; the worst results were produced in July due to the relatively higher number of sunny days in winter but more cloudy days, as well as complicated cloud types and higher aerosol concentrations in summer. Higher aerosol concentrations decrease the accuracy of the clear sky model. Different cloud types have different cloud microphysical properties, which include total cloud cover, cloud optical thickness, cloud equivalent temperature, liquid water column content in cloud, ice water column content in cloud, water particle radius, ice particle equivalent radius, etc. [53,54]. When there are more types of clouds and they change quickly, the GHI fluctuates considerably with no regularity, making the GHI difficult to forecast. In terms of the comparison of nRMSE values, the model using the FY-4A satellite images outperformed the SP model for all forecasting horizons and all seasons.

#### 4.1.2. Direct Normal Solar Irradiance (DNI)

The best results for DNI for both models were produced in October for all forecasting horizons, followed by January; the worst results were produced in July. The predicted results for DNI were significantly worse than for GHI for both models. The DNI prediction is very complicated; it is mainly related to the cloud moisture content, cloud height, cloud shape, dust in the atmosphere, and aerosol content. The daily fluctuation in the DNI component was often remarkable; the measured fluctuations in DNI ranged from a few hundred W/m^2^ to almost zero in minutes or seconds. The wide fluctuations in DNI mainly occurred due to its sensitivity to the variation in cloud cover. These steep fluctuations make predictions difficult. The model using FY-4A satellite images outperformed the SP model in July and October, with no improvement against the SP model in April, and slightly worse performance than the SP model in January.

#### 4.1.3. Skill Value for GHI and DNI

Based on Equation (12), the skill (SS) value of the SP model is zero. Figure 7 depicts the skill values for GHI and DNI for various forecast horizons in four typical months. Regardless of the forecast horizon and the season, the values of SS for GHI are all greater than zero. This indicated that the model using the FY-4A satellite images outperformed the SP model for all forecasting horizons and seasons; the best result was produced in October, with an SS value greater than 20%, followed by July, with an SS value greater than 10%. In terms of DNI, the SS of the FY-4A satellite model from high to low was July, October, April, and January. The skill value was greater than zero for all forecasting horizons in July and October, with values of about 12% and 11%, respectively. In April, the SS value was slightly negative. In January, the SS was negative for all forecasting horizons, increasing with increases in the forecasting horizon. This indicated that the model we used for DNI forecasting outperformed the SP model in July and October. The negative SS in January suggests that FY-4A forecasts are not as good as in the other three months.

### 4.2. Annual Performances

#### 4.2.1. Forecast against Ground Measurements: GHI

The performance of the algorithm was also be visualized using a scatter diagram. Figure 8 shows the GHI values forecasted by the model using FY-4A satellite observations versus ground measured values for the 30- to 180-min time horizon for the four months (October 2018; January, April, and July 2019), totalling about 4019 forecasting time points. The scatter diagram depicts the result of each forecast point. Table 1 shows the statistical indexes for GHI values for the 30–180 min time horizon.

The scatter diagrams show that the points are relatively concentrated, but become more scattered with the increase in time horizon. More discrete points appear in the peak and valley periods. The model using FY-4A satellite observations performed best at the 30-min time horizon; the value of nRMSE increases with increasing forecast horizon. Over time, the physical properties of the clouds are more likely to change. The nRMSE values for GHI for 30, 60, 90, 120, 150, and 180 min horizons were 26.78%, 31.51%, 33.74%, 34.73%, 35.7%2, and 36.84%, respectively; the nMAE values of GHI were 14.88%, 17.58%, 18.76%, 19.54%, 20.24%, and 21.35%, respectively. The GHI forecasting values were slightly underestimated (negative MBE and nMBE) for 30–150 min horizons and overestimated (positive MBE and nMBE) for the 180 min horizon. The SS value reached a maximum for the 30 min time horizon, with a value of 20.44% compared with the SP model. The skill values were all above 0 for all time horizons, confirming that the proposed algorithm is more accurate and reliable than the SP model. A similar conclusion was drawn from the GHI cumulative frequency curves (Figure 9). However, for the minimum and maximum value forecasts, the FY-4A model results were not as good as in other periods.

#### 4.2.2. Forecast against Ground Measurements: DNI

Figure 10 shows the DNI forecasting values versus ground measured values for the time horizon from 30 to 180 min for the four months. Figure 11 depicts the DNI cumulative frequency curves. Table 2 lists statistical indexes for the DNI value for the 30–180 min time horizon. The nRMSE values for DNI for the FY-4A satellite observation model for 30, 60, 90, 120, 150, and 180 min horizons were 40.96%, 46.98%, 50.03%, 53.04%, 55.81% and 58.80%, respectively; the DNI nMAE values were 25.33%, 29.34%, 31.50%, 33.59%, 35.72% and 38.15% respectively. The DNI forecasting values were overestimated (positive MBE and nMBE) for all horizons. The DNI skill (SS) values for the 30–180 min horizons were 6.20%, 0.40%, 3.99%, 5.05%, 6.47%, and 6.62%, respectively. The SS values were all above zero for all time horizons. Therefore, compared with the SP model, the FY-4A forecast model is slightly better.

The DNI prediction performance of the models was less satisfying than for GHI because DNI is very sensitive to meteorological conditions. We observed that the scatter distribution of DNI was more discrete than for GHI. The larger deviation occurred during steep fluctuation periods. Especially in summer, the type, height, and density of clouds change dramatically. Thick, low-altitude clouds, usually liquid water clouds, can cause the DNI to change from its maximum to zero in seconds. In addition, thin high-altitude clouds, usually composed of ice crystals, such as cirrus, when the solar elevation angle is above 10°–20°, DNI can pass through the thin high-altitude clouds. The aerosol content in summer in our study area is significantly higher than during other seasons. Aerosols account for a very small proportion of the mass in the atmosphere, but have a significant impact on radiation transfer and climate. The scattering coefficient, absorption coefficient, scattering phase function, and their variation with wavelength determine the direct radiative forcing of aerosols [55], but large uncertainties still exist in the estimation of direct radiative force using aerosol. At present, the lack of sufficient aerosol concentration observation data complicates the prediction of direct solar irradiance. So, DNI forecasting is more cmplicated and inaccurate than GHI. In our next study, we will validate the model under different climatic conditions.

## 5. Conclusions

In this paper, we presented an algorithm to forecast very short term (0–3 h) surface solar irradiance using FY-4A satellite observations. The algorithm obtains the cloud motion vector (CMV) using particle image velocimetry (PIV). When the cloud index forecast values in different time horizons were made, we obtained GHI and DNI forecast values in 15-min time resolutions for 30- to 180-min time horizons based on the Heliosat-2 method. The forecast results were validated using the data from the Chengde observation station for four typical months (October 2018; January, April, and July 2019). The smart persistence (SP) model was used as a reference model. The forecast results were analyzed using the statistical indexes (RMSE, MAE, MBE, nRMSE, nMAE, nMBE, *R*, and SS values), as well as a scatter diagram and cumulative frequency curves.

A seasonal study was conducted that demonstrated that forecasting in July and April is more difficult than in October and January. For GHI forecasting, the proposed model using FY-4A satellite images outperformed the SP model for all forecasting horizons and seasons, with the best results being produced in October. The SS value was above 20%. In terms of DNI forecasting, the results were worse than for GHI. The model we proposed for DNI forecasting outperformed the SP model in July and October; the skill values were about 12% and 11%, respectively.

The annual performance (four typical months in total) was evaluated with the measured values. The results showed that the nRMSE values of GHI for the 30–180 min horizon were 26.78–36.84%, and that the skill (SS) value reached a maximum for the 30 min time horizon, with a value of 20.44%. The skill values were all above zero for all time horizons, confirming that the proposed algorithm is more accurate and reliable than the SP model. In terms of DNI forecasting, we observed that larger deviations occurred during steep fluctuation periods. The SS values were all above zero for all time horizons. The SS reached a maximum at the 180-min time horizon, with a value of 6.62%.

The very short term (0–3 h) GHI forecasting approach was proven to be accurate and reliable. The DNI forecasting was slightly better than that of the SP model, because it is complicated and sensitive to local meteorological conditions. Therefore, in the future, we will verify the model in other regions under different meteorological conditions. It is worth noting that all relevant technologies have not been specifically developed for DNI prediction, whereas DNI forecasting is derived from the GHI forecasting model.

In the future, the model needs to be further improved. A high-precision algorithm for aerosol optical thickness should be developed using FY-4A satellite observations to improve the TL calculation accuracy to improve the clear sky model accuracy. This is important for modeling the process of cloud growth and dissipation in the future, which can improve the prediction accuracy with the forecast time horizon increasing. Surface solar irradiance forecasting under hazy weather conditions is also planned, because many haze processes occur in the Beijing-Tianjin-Hebei region, which will be more meaningful for the application of the model.

## Figures and Tables

**Figure 1 sensors-20-02606-f001:**
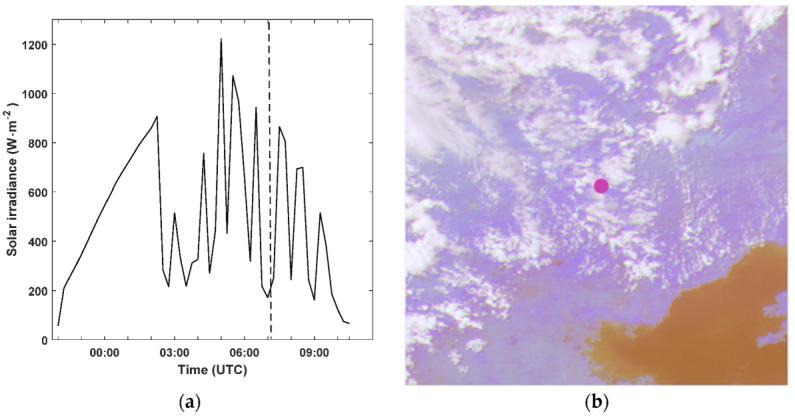
Example of variation curve of the global horizontal irradiance (GHI) measurement at meteorological station in Chengde on 1 July 2019 (**a**) and time-matched FY-4A satellite images on 1 July, 2019 at 07:15 UTC (**b**). The red point is the location of the ground observation station.

**Figure 2 sensors-20-02606-f002:**
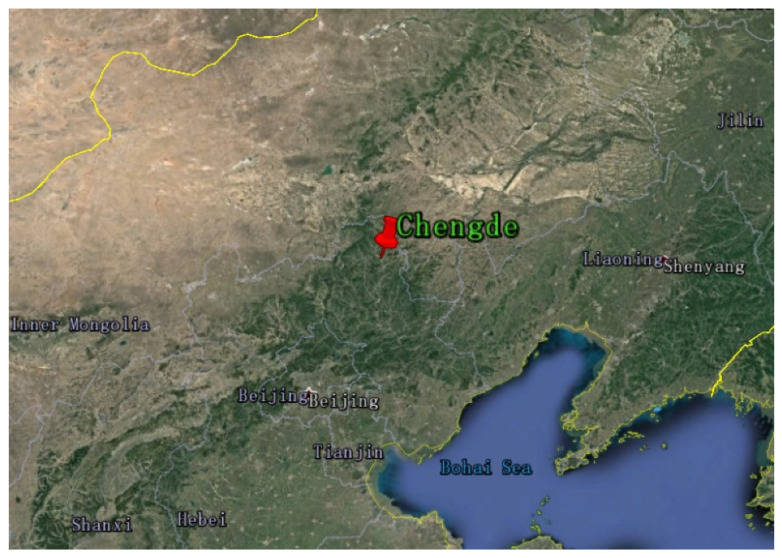
Location of the meteorological station of Chengde (from Google Earth).

**Figure 3 sensors-20-02606-f003:**
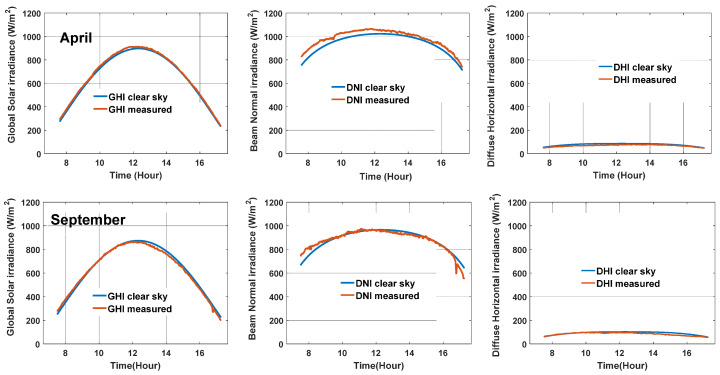
The ground measured and forecast results of GHI, DNI, and DHI under clear sky conditions for one day: 2 April (**top**) and 8 September (**bottom**), 2019.

**Figure 4 sensors-20-02606-f004:**
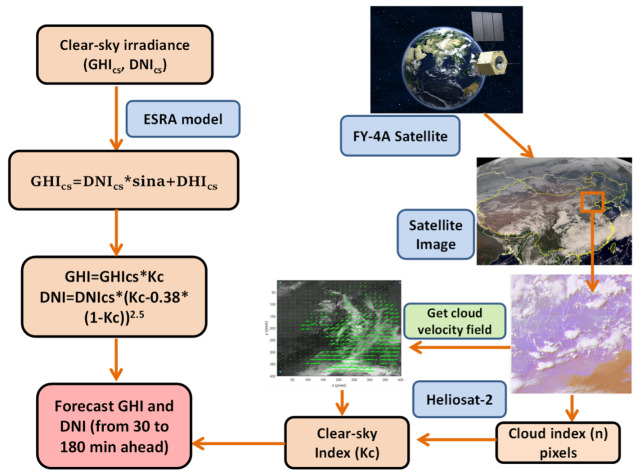
Overview of GHI and DNI forecast process with time horizons of 30 to 180 min.

**Figure 5 sensors-20-02606-f005:**
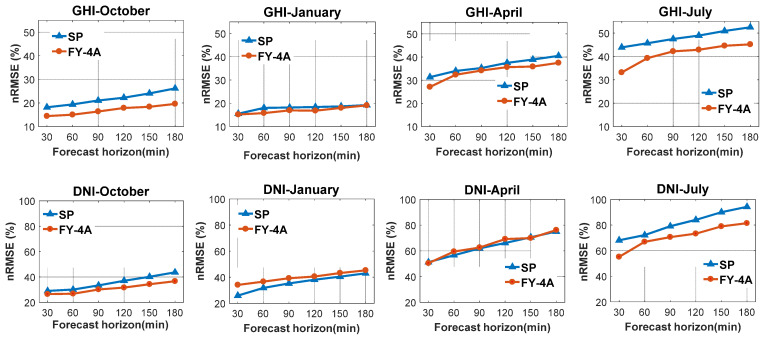
The nRMSE values of forecasting models for GHI (**top**) and DNI (**bottom**) in various forecast horizons in four typical months (October 2018; January, April, and July 2019).

**Figure 6 sensors-20-02606-f006:**
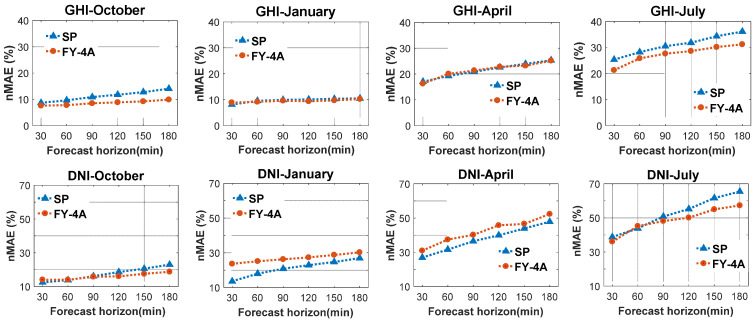
The nMAE values of forecasting models for GHI (**top**) and DNI (**bottom**) in various forecast horizon in four typical months (October 2018; January, April, and July 2019).

**Figure 7 sensors-20-02606-f007:**
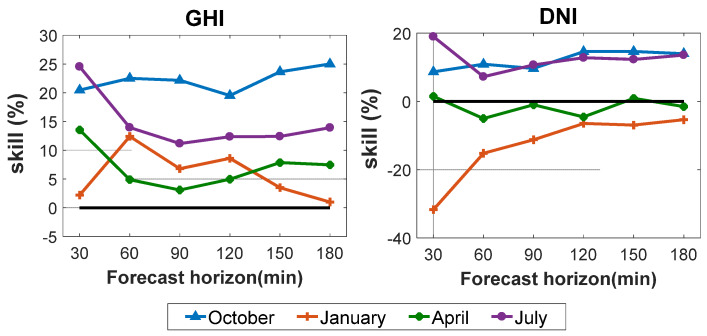
The skill values for GHI (**left**) and DNI (**right**) of various forecast horizons in four typical months (October 2018; January, April, and July 2019).

**Figure 8 sensors-20-02606-f008:**
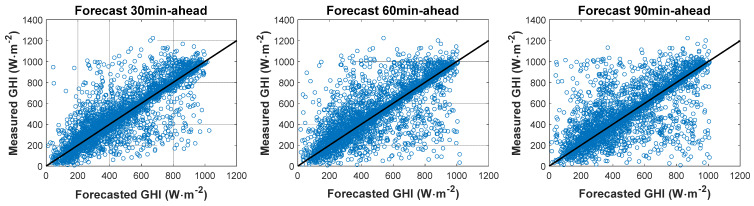
Forecasting GHI values by the model using FY-4A satellite observation versus ground measured values for time horizons from 30 to 180 min.

**Figure 9 sensors-20-02606-f009:**
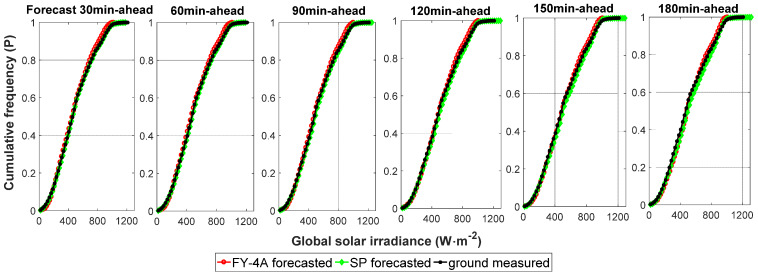
The GHI cumulative frequency curves for the time horizons from 30 to 180 min.

**Figure 10 sensors-20-02606-f010:**
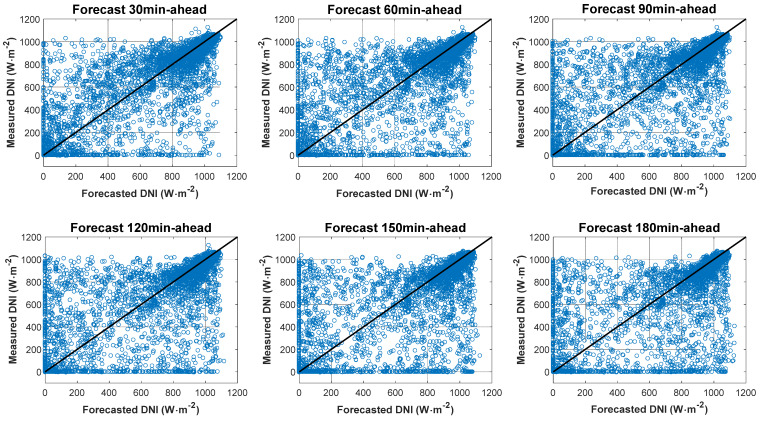
Forecasting DNI values by the model using FY-4A satellite observations versus ground measured values for the time horizon from 30 to 180 min.

**Figure 11 sensors-20-02606-f011:**
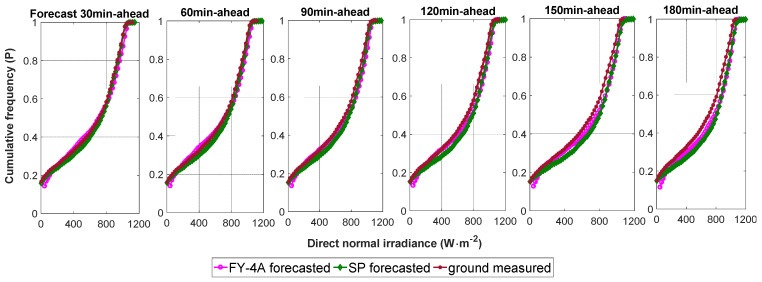
The DNI cumulative frequency curves for time horizons from 30 to 180 min.

**Table 1 sensors-20-02606-t001:** Statistical indexes for GHI value from 30 to 180 min time horizons (W/m^2^ for mean value, RMSE, MAE, and MBE).

Metric	Model	30 min	60 min	90 min	120 min	150 min	180 min
Mean value		493.94	504.67	511.89	516.60	517.68	514.11
RMSE	SP	166.09	180.93	190.30	200.08	209.11	215.73
	FY-4A	132.29	159.04	172.75	179.41	184.94	189.44
nRMSE (%)	SP	33.66	35.89	37.22	38.80	40.50	42.04
	FY-4A	26.78	31.51	33.74	34.73	35.72	36.84
MAE	SP	81.96	94.58	103.26	110.03	117.65	123.41
	FY-4A	73.54	88.75	96.07	100.95	104.81	109.79
nMAE (%)	SP	16.61	18.76	20.20	21.34	22.79	24.05
	FY-4A	14.88	17.58	18.76	19.54	20.24	21.35
MBE	SP	5.15	8.31	12.08	15.38	19.42	24.24
	FY-4A	−10.33	−10.19	−8.08	−5.16	−2.32	5.67
nMBE (%)	SP	1.04	1.65	2.36	2.98	3.76	4.72
	FY-4A	−2.09	−2.02	−1.57	−1.00	−0.45	1.10
*R*	SP	0.77	0.73	0.70	0.68	0.66	0.65
	FY-4A	0.85	0.78	0.74	0.73	0.72	0.71
SS (%)	FY-4A	20.44	12.20	9.35	10.49	11.80	12.37

**Table 2 sensors-20-02606-t002:** Statistical indexes for DNI value for 30–180 min time horizon of the four month (W/m^2^ for mean value, RMSE, MAE, and MBE).

Metric	Model	30 min	60 min	90 min	120 min	150 min	180 min
Mean value		586.00	592.98	594.57	594.17	592.17	588.47
RMSE	SP	255.55	284.52	308.61	331.11	352.76	370.41
	FY-4A	240.01	278.61	−296.76	315.33	330.51	346.05
nRMSE (%)	SP	43.67	47.17	52.11	55.86	59.67	62.97
	FY-4A	40.96	46.98	50.03	53.04	55.81	58.80
MAE	SP	130.71	156.67	178.39	196.53	215.94	231.97
	FY-4A	148.43	173.99	186.87	199.70	211.52	224.51
nMAE (%)	SP	22.34	26.53	30.12	33.16	36.53	39.43
	FY-4A	25.33	29.34	31.50	33.59	35.72	38.15
MBE	SP	20.02	27.96	38.14	47.85	59.69	73.37
	FY-4A	5.51	5.34	16.68	22.52	36.02	50.39
nMBE (%)	SP	3.42	4.73	6.44	8.07	10.10	12.47
	FY-4A	0.94	0.90	2.81	3.79	6.08	8.56
*R*	SP	0.77	0.72	0.67	0.62	0.58	0.54
	FY-4A	0.80	0.73	0.70	0.66	0.63	0.60
SS (%)	FY-4A	6.20	0.40	3.99	5.05	6.47	6.62

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
