# Peer review of "Very Short-Term Surface Solar Irradiance Forecasting Based On FengYun-4 Geostationary Satellite"

_sensors, 2020, doi:10.3390/s20092606_

Round 1

Reviewer 1 Report

Yang et al. described the algorithm of very short-term surface solar irradiance forecasting based on FY-4 satellite and evaluated the satellite product with ground-based measurements and smart persistence forecast. The authors have made forecasts for GHI and DNI. The results for GHI are good compared to the  ground-based measurements and the SP. The authors have included lots of recent publications in the introduction. It is an interesting paper and well-written.

Specific comments:

Line 70, in the citation, you only need to use the name of the first author, for example using Arbizu-Barrena et al., instead of listing five names. Similar for other citations in the paper.

Line 121, Why two channels of FY-4 are used in this study? How are they used?

Are they used in the calculation of CMV or cloud index or both? Please explain in the paper.

Fig. 3, Please give the date in the caption.

Line 191, Is Eq. 2 used for both the global irradiance and the direct irradiance?

Do you have a reference that Eq. 2 has been used to derive direct irradiance? In the CM-SAF solar radiation and surface albedo product, Eq. 2 is only used to calculate the global irradiance. The direct irradiance is calculated using a different formula. See the ATBD of the CM-SAF product  https://www.cmsaf.eu/SharedDocs/Literatur/document/2017/saf_cm_dwd_atbd_meteosat_hel_2_2_pdf.pdf?__blob=publicationFile

Eq. 3 Please give the values for the apparent albedo for the ground and clouds and explain how the apparent albedo values are derived.

3.3 Please explain the PIV method, give some references or websites.

Line 266, Provide the monthly mean AOD for each month in the paper.

In the tables, Could you give the mean values of GHI and DNI, respectively? You can add a column in the table or give the values in the text.

Reviewer 2 Report

The authors present a methodology for short-term forecasting solar irradiance by using geostationary satellite imagery. espite this topic has been already covered by several authors, the main novelty of this paper in my opinion is the development of the methodology applied to the Chinese Fen-Yung satellite. The paper is well-written and organised. I consider the paper worths to be published wih a few comments. I have only minor questions to the authors.

The forecasting method, PIV, is well-known but I think that the authors should described better this methodology and how they have implemented, since this is the core of the forecasting approach.

In Figure 10 the scatter plots show a lot of values  estimated when the experimental measurement of DNI is zero. The author should explain more in detail this; are they during night?, are they forecasted values under very cloudy situations. They should put the discussion into the context

Reviewer 3 Report

Referees synopsis
The authors present a method to forecast GHI and DNI using geostationary satellite data. The forecast results are compared to station data and a persistence model. For GHI the satellite-based model outperfoms the persistence model, while the performance for DNI is less good.
The manuscript is well structured and written and easy to understand for the reader. The scientific added value is limited as this study applies a simple approach and the results leave lots of room for improvements. This study has the potential to be a useful and important one, but at this point it feels to be not ready for publication. There are several points, which should be improved before publication. Nevertheless, this topic might be of interest for applications in the field of solar energy and so far there are few applications of FY4 data in this field.

Major review points
1. The authors mention several more sophisticated methods using Meteosat data in the introduction. Regarding the poor results of the forecast it should considered using one of these to demonstrate the possible gain in performance compared to the FY-4A method.
2. In the conclusion it is stated that ‘the model needs to be further improved’. That’s my impression too. This publication would definitely be of higher significance if the results would be improve before publication.
3. The conclusion also states that the authors will further verify the model in other regions with different meteorological conditions. I support this idea, but maybe this should be done already within this study.

Minor review points
1. Line 51-53: Very complicated sentence, which should be rephrased.
2. Line 70: Why not using et al.?
3. Line 103: the SP model is used a lot in this study and needs a little more introduction.
4. Line 108: Why is ‘lower computaional complexity’ needed in this case. Please explain in more detail.
5. Line 108: What does ‘higher efficiency’ mean in this framework?
6. Section 4.1.3: The contradiction between a high skill and high errors at the same time might be confusing to some readers and should be explained in more detail. Is a higher skill really good if there is a high error at the same time?
7. Line 326: Please explain cumulative frequency curves in more detail.

Round 2

Reviewer 3 Report

The authors have implemented each of the reviewer comments, which improved the quality of the mansucript significantly.